# Urban Particulate Matter Enhances ROS/IL-6/COX-II Production by Inhibiting MicroRNA-137 in Synovial Fibroblast of Rheumatoid Arthritis

**DOI:** 10.3390/cells9061378

**Published:** 2020-06-02

**Authors:** Ming-Horng Tsai, Miao-Ching Chi, Jen-Fu Hsu, I-Ta Lee, Ko-Ming Lin, Mei-Ling Fang, Ming-Hsueh Lee, Chiang-Wen Lee, Ju-Fang Liu

**Affiliations:** 1Department of Pediatrics, Division of Neonatology and Pediatric Hematology/Oncology, Chang Gung Memorial Hospital, Yunlin 638, Taiwan; mingmin.tw@yahoo.com.tw; 2College of Medicine, Chang Gung University, Taoyuan 333, Taiwan; 3Chronic Disease and Health Promotion Research Center, Chang Gung University of Science and Technology, Puzi City, Chiayi County 613, Taiwan; mcchi@mail.cgust.edu.tw; 4Division of Pulmonary and Critical Care Medicine, Chiayi Chang Gung Memorial Hospital, Puzi City, Chiayi County 613, Taiwan; 5Department of Respiratory Care, Chang Gung University of Science and Technology, Puzi City, Chiayi County 613, Taiwan; 6Department of Pediatrics, Division of Neonatology, Chang Gung Memorial Hospital, Lin-Kou, New Taipei City 333, Taiwan; jeff0724@gmail.com; 7School of Dentistry, College of Oral Medicine, Taipei Medical University, Taipei City 111, Taiwan; itlee0128@tmu.edu.tw; 8Division of Allergy, Immunology and Rheumatology, Department of Internal Medicine, Chang Gung Memorial Hospital, Puzi City, Chiayi County 613, Taiwan; koming@cgmh.org.tw; 9Center for Environmental Toxin and Emerging-Contaminant Research, Cheng Shiu University, Kaohsiung 83347, Taiwan; k6764@gcloud.csu.edu.tw; 10Super Micro Research and Technology Center, Cheng Shiu University, Kaohsiung 83347, Taiwan; 11Division of Neurosurgery, Department of Surgery, Chang Gung Memorial Hospital, Chia-Yi 61363, Taiwan; ma2072@gmail.com; 12Department of Orthopaedic Surgery, Chang Gung Memorial Hospital, Puzi City, Chiayi County 61363, Taiwan; 13Department of Nursing, Division of Basic Medical Sciences, and Chronic Diseases and Health Promotion Research Center, Chang Gung University of Science and Technology, Puzi City, Chiayi County 61363, Taiwan; 14Department of Safety Health and Environmental Engineering, Ming Chi University of Technology, New Taipei City 24301, Taiwan; 15College of Medicine, Chang Gung University, Guishan Dist, Taoyuan City 33303, Taiwan; 16Translational Medicine Center, Shin-Kong Wu Ho-Su Memorial Hospital, Taipei City 11101, Taiwan; 17Department of Medical Research, China Medical University Hospital, China Medical University, Taichung 40402, Taiwan; 18School of Oral Hygiene, College of Oral Medicine, Taipei Medical University, Taipei City 11031, Taiwan

**Keywords:** particulate matter, air pollution, rheumatoid arthritis, reactive oxidative stress, interleukin-6, inflammation, MAPK signaling

## Abstract

Background: Rheumatoid arthritis (RA) has been associated with air pollution, possibly due to the augmentation of inflammatory effects. In this study, we aimed to determine the roles of inflammatory pathways and microRNA involved in the pathogenesis of RA fibroblast-like synoviocytes (FLS) inflammation induced by particulate matter. Methods: The inflammatory mediators, messenger RNAs, microRNAs and their interrelationships were investigated using western blotting, QPCR, ELISA and immunohistochemistry. Results: Particulate matter (PMs) induced an increase in the expression of interleukin-6 (IL-6) and cyclooxygenase-II (COX-II) in RA-FLS and microRNA-137 was found definitely to mediate the inflammatory pathways. PMs-induced generation of reactive oxygen species (ROS) in RA-FLS was attenuated by pretreatment with antioxidants. Nox-dependent ROS generation led to phosphorylation of ERK1/2, p38 and JNK, followed by downregulation of microRNA-137. In vivo studies, the joints of rats exposed to PMs revealed synovial fibroblast inflammation under pathologic examination and the expressions of IL-6 and COX-II were obviously increased. PMs exposure results in activated ROS-mediated mitogen-activated protein kinase (MAPK) signaling pathways and cause increased IL-6 and COX-II through downregulation of hsa-miRNA-137, which lead to inflammation and RA exacerbation. Conclusions: microRNA-137 plays an important role in PMs-induced RA acute exacerbation through MAPK signaling pathways and IL-6/COX-II activation. Targeting these mechanisms can potentially be used to develop new therapeutic strategies and prevention of RA inflammation in the future.

## 1. Introduction

Rheumatoid arthritis (RA) is a systemic autoimmune disease characterized by chronic inflammation and progressive damage to joints. If joint damages are untreated, morbidity and mortality will increase [1,2]. The pathogeneses of RA include infiltration of various inflammatory cells and a crosstalk with cytokines, including IL-6, IL-17, TNF-α and GM-CSF that mediate the immune responses and cause disease onset, persistence and subsequent joint destruction [3,4]. Currently, a combination of genetic and environmental factors that eventually converge in the over-reactive immune system is considered as the basic concept of RA etiology [5,6]. Several genes were identified to contribute to RA susceptibility and epigenetic modulations as well as gene mutation are considered to be associated with RA disease presentation [7,8].

Environmental factors contribute to the onset of autoimmune diseases. One of the important environmental contributors is air pollution, which usually increases with the development of modern countries [9,10]. Several epidemiological studies and experimental models have demonstrated the association between particulate matter (PMs), smoking and certain occupational exposure and RA exacerbation [10,11,12,13]. Of particular concern is the ambient fine particulate matter (PMs), a mixture of multiple components (such as metals, water-soluble ions, polycyclic aromatic hydrocarbons [PAHs] and organic carbons, etc.) and diversified sources (such as coal combustion, ground dust, biomass burning and vehicle exhaust) with high oxidative potential and small aerodynamic diameter (less than 2.5 micrometers) that can penetrate deep into the pulmonary microvascular system and affect various organ systems [14,15]. PMs are found to cause detrimental inflammation on various target organs through producing reactive oxygen species (ROS) [16,17,18]. The oxidative effects of PMs on pulmonary epithelial cells are mediated through pro-inflammatory cytokines such as interleukin-6 (IL-6), IL-8 and cyclooxygenase II (COX-II), which are also involved in the pathogenesis of RA exacerbation [19,20,21]. Numerous studies have elucidated the potential biologic effects of PMs, including induction of acute inflammation, influencing adaptive immune responses and a significant role for epigenetic mechanisms. We have previously demonstrated that PMs can induce inflammation in human fibroblast-like synoviocytes (FLSs) [22], which mimic the pathogenesis of RA; however, little is known about the link between inflammatory pathway and epigenetic changes in RA patients after PMs exposure.

MicroRNAs (miRNAs) are small, noncoding RNAs that regulate expression of functionally related protein-coding genes and act as key regulators of inflammation [23,24]. Under external stimulation or oxidative stress, miRNA expression can be upregulated or downregulated, which may influence disease presentation [25]. The epigenetic effects of miRNAs were demonstrated to regulate FLS proliferation in RA patients, contribute to or ameliorate RA disease progression and can be the biomarker for RA diagnosis [26,27,28,29]. Other studies have also found that miRNA profiles in blood were altered by air pollution exposure and these changes were associated with immune responses and oxidative stress in chronic inflammatory diseases [30,31,32]. We hypothesized that exposure to PMs would alter miRNA expression in the FLS, leading to exacerbation of acute joint inflammation. In this study, we sought to determine whether ambient PMs is associated with the expression of specific miRNA and to investigate the participation of such miRNA in the inflammatory pathways related to RA.

## 2. Materials and Methods

### 2.1. Materials

Synthesized sequences of a human miRNA137 (hsa-miR-137) mimic, a hsa-miR-137 inhibitor and a negative control miRNA were purchased from GeneDireX (Las Vegas, NV). U0126, SB203580, SP600125, PD98059, N-acetylcysteine (NAC), Apocynin (APO), Diphenyleneiodonium (DPI) and catalase were supplied by Sigma-Aldrich (St Louis, MO, USA). The MEK1 dominant-negative mutant was a gift from Dr. W.M. Fu (National Taiwan University, Taipei, Taiwan). The ERK2 (K52R) dominant-negative mutant was a gift from Dr. M. Cobb (Southwestern Medical Center, Dallas, TX, USA). The p38 dominant negative mutant was provided by Dr. J. Han (Southwestern Medical Center, Dallas, TX). The JNK dominant negative mutant was provided by Dr. M. Karin (University of California, San Diego, CA, USA). The human IL-6 promoter construct pIL6-luc651 (-651/+1) was gifts from Dr. Oliver Eickelberg (Ludwig Maximilians University Munich, Munich, Germany). The human COX-II promoter construct pCOX-2-Luc (−891/+9) was gifts from Dr. Jin-Ching Lee (Kaohsiung Medical University, Kaohsiung, Taiwan).

Cell culture supplements were purchased from Invitrogen (Carlsbad, CA, USA). Luciferase^®^ Reporter Assay System was bought from Promega (Madison, WI, USA). All other chemicals not described above were supplied by Sigma-Aldrich (St Louis, MO, USA).

### 2.2. Human Synovial Fluids and Tissues

Human synovial fibroblasts (SFs) were isolated by collagenase treatment of synovial tissue samples obtained from 12 patients with RA undergoing knee replacement surgery in Chiayi Chang Gung Memorial Hospital (CCGMH). In addition, 12 samples of nonarthritic synovial tissues obtained arthroscopy after trauma joint derangement were used as the control. Among the 12 RA patients, 7 (58.3%) were under steroid treatment and all patients had DMARDs at the time of enrollment. The DMARDs included methotrexate (7 patients, 58.3%), hydroxychloroquine (3 patients, 25%) and sulfasalazine (2 patients, 16.7%). Written informed consent was obtained from all patients recruited into this study, and this study was approved by the Institutional Review Board of CCGMH. The certificate of the IRB number was 201600517A3. The synovial tissue was obtained from patients with RA and the controls. Fresh synovial tissues were finely minced and digested in Dulbecco’s modified Eagle’s medium (DMEM) containing 2 mg/mL type II collagenase (Sigma-Aldrich, St. Louis, MO, USA) for 4 h at 37 °C and under 5% CO_2_. Passages 5–7 of the obtained RASFs were used in this study. Results of four independent experiments were presented.

Patients were included if they meet the following criteria:(1)Patients who were able and willing to provide written informed consent;(2)Patients who had sufficient knowledge to understand Chinese or Taiwanese language, so they could comply with the requirements of the study;(3)Patients who were at least 18 years old, but less than 90 years old;(4)Patients who were diagnosed as having RA by their rheumatologist and met the 2010 ACR/EULAR classification criteria for RA (Aletaha D, et al., 2010);(5)Patients who had active RA defined as a clinical disease activity index (CDAI) > 10 and had a swollen knee joint;

The exclusion criteria of this study were:(1)Patients who had any other inflammatory rheumatic disease than RA, including secondary Sjögren’s syndrome;(2)Patients who refused to sign or who could not understand either Chinese or Taiwanese;(3)Patients who could not tolerate the procedure of knee treatment.(4)Patients who were pregnant.

Human fibroblast-like synoviocytes (H-FLS) were obtained from Cell Applications INC (USA) and human rheumatoid arthritis fibroblast-like synoviocytes cell line (MH7A) were obtained from Riken cell bank (Ibaraki, Japan). Cells were cultured in the Roswell park memorial institute (RPMI)-1640 (Wako, Osaka, Japan) supplemented with 10% heat-inactivated fetal bovine serum (FBS) (Gibco, Eggenstein, Germany), penicillin (final concentration, 100 U/mL) and streptomycin (final concentration, 0.1 mg/mL) in a humidified atmosphere of 5% CO_2_ and 95% air at 37 °C.

### 2.3. Preparation of Particle Matter Samples

The particle matters (PMs) (SRM 1649b, obtained from NIST; Gaithersburg, MD, USA) were prepared at a concentration of 1000 μg/mL in PBS. The suspended particles were then sonicated for 30 min to avoid agglomeration.

For other experimental designs and procedures, including cell cytotoxicity assay, cell cycle analysis, quantification of mRNA and miRNA by real-time quantitative polymerase chain reaction amplification, enzyme-linked immunosorbent assay, transfection and reporter gene assay, western blot analysis, measurement of intracellular ROS accumulation, predicted and validated microRNA target interactions and plasmid construction and luciferase assay, please see the Appendix A.

### 2.4. Transient Transfection

Cells were transfected with hsa-miR-137 mimic, control mimic, vector, dominant negative MEK mutants, dominant negative ERK mutants, dominant negative JNK mutants and dominant negative p38 mutants and luciferase plasmid by using Lipofectamine 3000 in culture medium. After 24 h of transfection, cells were incubated with the indicated agents. After 24 h of incubation, the luciferase activity in the transfected cells was measured using a luciferase reporter assay system (Promega) according to the manufacturer’s instructions. Transactivation was determined by monitoring the firefly luciferase levels in the pGL2 vector. The luciferase assay was performed by adding lysis buffer (100 μL) and harvesting the cells through centrifugation (13,000 rpm for 5 min). The supernatant was transferred to fresh tubes and 20 μL of cell lysate was added to 80 μL of fresh luciferase assay buffer in an assay tube. The luciferase activity was measured using a microplate luminometer. Luciferase activity was normalized to transfection efficiency based on the cotransfected β-galactosidase expression vector.

### 2.5. Reporter Gene Assay

Human synovial fibroblasts were co-transfected with 0.8 μg IL-6 or COX-II luciferase plasmid and 0.4 μg β-galactosidase expression vector. Fibroblasts were grown to 80% confluent in 12 well plates and were transfected the following day with Lipofectamine 3000 (LF3000; Invitrogen). DNA and LF3000 were premixed for 20 min and then applied to cells. After 24 h transfection, cells were then incubated with the indicated agents. After further 24 h incubation, the media were removed and cells were washed once with cold PBS. To prepare lysates, 100 μl reporter lysis buffer (Promega, Madison, WI, USA) was added to each well and cells were scraped from dishes. The supernatant was collected after centrifugation at 13,000 rpm for 10 min. Aliquots of cell lysates (20 μl) containing equal amounts of protein (20–30 μg) were placed into wells of an opaque black 96-well microplate. An equal volume of luciferase substrate was added to all samples and luminescence was measured in a microplate luminometer. The value of luciferase activity was normalized to transfection efficiency monitored by the co-transfected β-galactosidase expression vector.

### 2.6. Plasmid Construction and Luciferase Assays

The 3′UTR-luciferase reporter constructs containing 3′UTR regions of IL-6 or COX-II with wild type and mutant binding sites of hsa-miR-137 were amplified by PCR method, cDNAs obtained from H293 T cells. PCR products were cloned into pmirGLO reporter vector (Promega) between PmeI and XbaI sites, instantly downstream of the luciferase gene. Mutant 3′UTR were constructed by introducing mismatched mutations into putative seed regions of IL-6 or COX-II, with all constructs containing 3′UTR inserts sequenced and verified. These plasmids with 3′UTR of IL-6 or COX-II and β-galactosidase as control were transfected into cells using lipofectamine 3000. Following transfection, these cells were incubated with the indicated agents. Cell extracts were prepared and used for measuring the luciferase and β-galactosidase activities as the manufacturer’s recommendations. Activities of luciferase and β-galactosidase were then measured by Luciferase Assay System (Promega, WI, USA).

### 2.7. Collagen-Induced Arthritis (CIA) Rat Model

Female Lewis rats, weighing between 175 and 200 g from Lasco (Taipei, Taiwan), were used in these studies. Subjects were randomized into the study groups, such that all groups had similar average baseline body weights and activity profiles prior to induction. We employed the CIA model, a well-established and validated rodent rheumatology model. Briefly, to induce arthritis, rats were anesthetized under isoflurane and injected intradermally with 2 mg/mL of porcine type II collagen (Chondrex; Redmond, WA, USA) in Incomplete Freund’s Adjuvant (IFA) (Sigma; St. Louis, MO, USA) at two sites on the back of the animal. A booster was given 7 days post-induction. Control animals were injected with IFA only. 32 rats were randomly assigned to 4 groups: 8 in the control group, 8 in the PMs-exposure group, 8 in the CIA group and 8 in the CIA-with-PMs-exposure group. Rats in the PMs-exposure group received PMs dose (4 mg/kg) with intratracheal instillation at day 21. Rats in the control group received an equivalent volume of normal saline with intratracheal instillation on the same days. The clinical severity of arthritis in each knee was measured in a blinded manner with a plethysmometer (Marsap, Mumbai, India, http://www.marsap.com) once weekly for 1 week. The rats were sacrificed on day 42 and the phalanges and knee joints were removed immediately and fixed in 4% paraformaldehyde for immunohistochemistry.

### 2.8. Immunohistochemistry

Rat joint tissue was fixed in 4% paraformaldehyde, decalcified in EDTA bone decalcifier and embedded in paraffin. The sections (7 μm) were stained with hematoxylin and eosin (H&E) and toluidine blue to detect proteoglycans. For immunohistochemistry, rat joint sections were blocked with 1% normal goat serum and stained with antibodies to IL-6 (1 μg/mL, Santa Cruz, CA, USA), COX-II (200 ng/mL, Santa Cruz, CA, USA) and isotype control antibody (1 μg/mL, Santa Cruz, CA, USA) at 4 °C overnight. After three washes in PBS, the secondary antibody (biotin-labeled goat anti-rabbit IgG) was applied for 1 h at room temperature. Staining was detected with 3, 3′-diaminobenzidine tetrahydrochloride and the sections were then counterstained with H&E and observed under a light microscope.

### 2.9. Statistical Analysis

The data were expressed as the mean ± SD. Statistical analysis was performed using SigmaStat 3.0 software. One-way analysis of variance (ANOVA) with Fisher’s LSD post hoc tests was used for statistical comparisons of more than two groups. In all cases, *p* < 0.05 was considered to be statistically significant.

## 3. Results

### 3.1. PMs Promote IL-6 and COX-II Expression in Human Rheumatoid Arthritis Fibroblast-Like Synoviocytes (RA-FLS)

We detected mRNA expression levels of inflammatory cytokines in RA-FLS. The cells were treated with PMs (50 μg/cm^2^) for 24 h. The mRNA expression of inflammatory cytokines were examined using qPCR. Moreover, then the expression of IL-6 and COX-II were significantly higher than that of other inflammatory cytokines in RA-FLS compared with the basal level expressed in controls (Figure 1A). To understand the relationship between PMs and IL-6 and COX-II in RA-FLS, we examined the level of IL-6 and COX-II after PMs treatment. RA-FLS were incubated with various concentrations of PMs for 24 h or PMs (50 μg/cm^2^) for 6, 12 or 24 h. The mRNA and protein expression of IL-6 and COX-2 were examined using qPCR and western blot. We found that PMs induced IL-6 and COX-II production in a concentration-dependent manner (Figure 1B,C) and induction occurred in a time-dependent manner in RA-FLS (Figure 1D). Furthermore, RA-FLS were incubated with various concentrations of PMs for 24 h; supernatants and cell lysates were then collected. The IL-6 level in the culture media was measured using a Quantikine ELISA kit. We observed that PMs markedly induced IL-6 release in a concentration-dependent manner according to ELISA analysis (Figure 1E). In addition, RA-FLS were incubated with various concentrations of PMs for 24 h. IL-6 and COX-2 luciferase activity was measured and the results were normalized to the β-galactosidase activity. PMs-induced IL-6 and COX-II expression significantly increased within 6 h and continued to increase over 24 h. In addition, PMs also induced the expression of IL-6 and COX-II promoter activity in these cells (Figure 1F,G). Taken together, these results suggest that PMs induces IL-6 and COX-II upregulation in human RA-FLS.

### 3.2. PMs Induces IL-6 and COX-II Expression via ROS

Several studies have demonstrated that reactive oxygen species (ROS) contributes to IL-6 and COX-II expression in various cell types. Thus, the role of ROS generation associated with IL-6 and COX-II expression in response to PMs was investigated. To confirm that the generation of ROS was involved in PMs-induced IL-6 and COX-II expression in RA-FLS, CellROX green reagent was used to measure the generation of ROS in these cells. The cells were labeled with CellROX green reagent and then treated with 50 μg/cm^2^ PMs and the resultant fluorescent intensity was measured at 485 nm excitation and 520 nm emission.

As shown in Figure 2A,B, the results indicated that PMs treatment of RA-FLS induced ROS accumulation. Moreover, pretreatment with N-acetylcysteine 1-mM (NAC, NADPH oxidase inhibitor), diphenyleneiodonium chloride 10 μM (DPI, nonspecific flavoprotein inhibitor), catalase 500 units/mL (ROS scavengers) and apocynin 100 μM (APO, NOX-like enzymes inhibitor) markedly inhibited PMs-induced ROS generation (Figure 2C). NAC, DPI, catalase and APO significantly abrogated PMs-induced IL-6 and COX-II protein and mRNA levels and promoter activity (Figure 2D–F). These results indicated that ROS generation plays a critical role in PMs-induced IL-6 and COX-II expression in human RA-FLS.

### 3.3. PMs Induce IL-6 and COX-II Expression via MAPK in Human RA-FLS

There are studies indicating that MAPK family members (ERK1/2, p38 MAPK and JNK1/2) play a role in IL-6 and COX-II gene expression [33,34]. Therefore, we examined whether MEK, ERK, JNK and p38 MAPK are also important mediators in PMs-induced IL-6 and COX-II expression in human RA-FLS. We found that the pretreatment with the inhibitor of ERK (PD98059; 20 μM), p38 MAPK (SB203580; 20 μM), JNK (SP600125; 20 μM) or MEK (U0126; 20 μM) inhibited enhancement of IL-6 and COX-II mRNA expression and promoter activity by PMs (Figure 3A,B). To further verify that PMs-induced IL-6 and COX-II expression was mediated via MAPK, cells were transfected with MEK, ERK, JNK and p38 MAPK mutants, and then incubated with PMs for 24 h. Transfection with MEK, ERK, JNK and p38 MAPK mutants markedly inhibited PMs-induced IL-6 and COX-II mRNA levels in RA-FLS (Figure 3C).

To determine whether MEK, ERK, JNK and p38 MAPK were activated during PMs-triggered IL-6 and COX-2 expression, we found increased phosphorylation of MEK, ERK, JNK and p38 MAPK in a time-dependent manner following PMs stimulation (Figure 3D). Moreover, we confirmed a relationship between ROS production and MAPK signaling pathways in response to PMs. Incubating the cells with the ROS inhibitor reduced PMs-induced increases in MEK, ERK, JNK and p38 MAPK phosphorylation (Figure 3E). These results indicated that MEK, ERK, JNK and p38 MAPK play critical roles in PMs-induced IL-6 and COX-II expression in human RA-FLS.

### 3.4. PMs Enhances IL-6 and COX-II Expression by Inhibiting Hsa-miR-137 Synthesis

The role of microRNAs demonstrate differential expression patterns between RA and heath people and are involved in the pathogenesis of RA [35,36]. This study identified that PMs promotes inflammation via IL-6 and COX-II. Next, we sought to determine whether specific miRNAs are involved in PMs-induced inhibition of IL-6 and COX-II expression. analysis of miRNA target prediction programs (miRsystem) confirmed that hsa-miR-137 directly targets the 3′-UTR region of IL-6 and COX-II (Figure 4A). hsa-miR-137 is located on human chromosome 1p22 and has been implicated to act as a suppressor in several inflammatory diseases. hsa-miR-137 that could possibly bind to the 3′-UTR region of IL-6 and COX-II mRNA, levels of hsa-miR-137 expression were significantly decreased by the greatest extent after PMs administration. To confirm these findings, we compared levels of hsa-miR-137 expression in RA-FLS treated with PMs 10–50 μg/cm^2^. PMs concentration- and time-dependently inhibited miR-137 expression (Figure 4B,C). We used the fluorescent dye-labeled miRNA mimic control and miRNA inhibitor control that can be easily observed under fluorescent microscope to determine the transfection efficiency. In our study, we demonstrate that up to 50% transfection efficiency can be obtained. (Figure 4D). To further determine whether PMs stimulates IL-6 and COX-II expression by inhibiting hsa-miR-137 synthesis, we transfected RA-FLS with hsa-miR-137 mimic and observed reductions in PMs-enhanced IL-6 and COX-II mRNA and protein secretion (Figure 4D–F). In the other hand, we also we transfected RA-FLS with hsa-miR-137 inhibitor and observed inductions in PMs-enhanced IL-6 and COX-II mRNA expression (Figure 4G). Furthermore, Pretreatment of cells with ROS inhibitors significantly reversed PMs-induced inhibition of hsa-miR-137 expression (Figure 4H). In addition, pretreated cells with MAPK inhibitors or transfected cells with MEK, ERK, JNK and p38 MAPK mutants significantly reversed PMs-induced inhibition of hsa-miR-137 expression (Figure 4I,J).

We also used the luciferase reporter vector, including the wild-type 3′UTR of IL-6 and COX-II mRNA (IL-6–3′UTR-WT and COX-II-3′UTR-WT) and the mutated vector harboring mismatches in the predicted hsa-miR-137 binding site (IL-6–3′UTR-MUT and COX-II-3′UTR-MUT), to determine whether hsa-miR-137 regulates transcription of the IL-6 and COX-II gene (Figure 5A). hsa-miR-137 mimic reduced PMs-enhanced luciferase activity in the IL-6–3′UTR-WT and COX-II-3′UTR-WT plasmid, but not in the IL-6–3′UTR-MUT and COX-II-3′UTR-MUT plasmid (Figure 5B,C). In addition, treatment with ROS inhibitors, MEK, ERK, JNK and p38 inhibitors or dominant-negative mutants reversed PMs-mediated IL-6–3′-UTR and COX-2–3′-UTR luciferase activity (Figure 5D–F). Collectively, these data suggest that hsa-miR-137 directly represses IL-6 and COX-II expression via binding to the 3′-UTR region of the human IL-6 and COX-II gene through the MAPK signaling pathway. Therefore, the present study aimed to determine whether hsa-miR-137 effects inflammation in RA and the results revealed that the overexpression of hsa-miR-137 substantially decreased the expression of IL-6 and COX-II in RA-FLS, suggesting that hsa-miR-137 may have potential as an anti-inflammatory therapy for RA.

### 3.5. PMs Enhanced CIA-Induced Arthritis

To further confirm the role of PMs in vivo, we first assessed the deteriorating effects of PMs on rats with CIA. Compared with controls, paw swelling was significantly prominent in CIA rat and worsened after administration of PMs (Figure 6A,B). IHC staining revealed substantially higher IL-6 and COX-II expression in CIA rats compared with controls. Additionally, treatment with PMs also enhanced IL-6 and COX-II expression over synovium specimens (Figure 6C). These results indicated that PMs augments disease activity in the CIA rat model.

## 4. Discussion

Few articles have explored the relationship between intra-articular inflammation and air pollution exposure in humans. However, it is increasingly important to understand the mechanisms by which air pollution induces the development and exacerbation of RA [37,38]. Because many epidemiological studies have suggested the relationship between air pollution and RA exacerbation [9,10,11,37], understanding these interactions may contribute to new preventative and therapeutic strategies. In this study, we found that high levels of IL-6 and COX-II in human RA-FLS are associated with PMs-induced suppression of hsa-miR-137, which influence RA-FLS inflammation. Specifically, PMs induces IL-6 and COX-II expression via MEK, ERK, JNK and p38 MAPK signaling pathways, as well as through downregulation of hsa-miR-137 expression.

miRNAs represent an important mode of epigenetic control in RA gene expression [5,6,25,26]. These noncoding RNAs are implicated in many biologic processes and play an important role in the inflammatory pathways. Previous studies have identified several miRNAs that can upregulate or downregulate RA-FLS [26,27,39,40,41,42], through various effects, such as Treg/Th17 balance, NF-kappaB regulatory circuit, TLR4-dependent cytokine release, affecting lymphocyte function and targeting other inflammatory mediators. In the current study, we first identified hsa-miR-137 as the potential regulator of IL-6 and COX-II expression through the predictive software. hsa-miR-137 has previously been documented as an important regulator of cancer development by targeting many different mRNAs to inhibit the release of inflammatory cytokines, proliferation and migration of several cancer cells [43,44,45]. A recent study also found that hsa-miR-137 can decrease proliferation, migration and invasion of RA-FLS [46]. We then found downregulation of hsa-miR-137 after PMs exposure in RA-FLS, suggesting its role in the PMs-induced RA exacerbation. Our data also showed that overexpression of hsa-miR-137 is associated with decreased IL-6 and COX-II in RA-FLS. Therefore, the IL-6/COX-II-miR-137 axis may represent a novel pathogenesis of RA-FLS autoimmune inflammation and hsa-miR-137 can be a novel therapeutic target for the treatment of RA exacerbation.

In the current study, we demonstrated for the first time that IL-6 is the pro-inflammatory cytokine and IL-6 and COX-II activation is the main mediators that trigger RA after exposure to air pollution. Serum level of IL-6 has a significant relationship with RA severity and disease activity [47,48]. IL-6 is involved in the pathogenesis of RA exacerbation through various signal pathways, including STAT-1 and STAT-3 recruitment and activation, pathogenic Th17/protective Treg imbalance and phosphorylation of tyrosine kinase in JAK family [19,20,21]. Targeting the IL-6 receptors was shown effective in decreasing the RA disease activity, reducing the bone erosion and improving the disability scores [47,48]. In 2020, Xu et al., provided evidence that PM2.5 exposure could activate oxidative stress-JAK2/STAT3 signaling pathway, elevate the levels of IL-6, IL-8 and COX-2 in human bronchial epithelial cells [49]. Our previous study showed that PMs induce COX-2 in RA-FLS [22]. Air pollution is an important global issue and the potential determinant. Our study suggested that the targeting IL-6 and COX-2 may attenuate PMs-induced arthritis.

In contrast to previous studies that have found Th17/Tregs imbalance and STAT activation as the key mechanism in IL-6 associated RA immunopathology [19,20,21], we found that MAPK family members (ERK1/2, p38 MAPK and JNK1/2) are the major signaling pathways after PMs exposure. We found that inhibition of hsa-miR-137 will activate a luciferase reporter constructs containing the IL-6 and COX-II 3′UTRs and these effects can be reversed by inhibitors of ROS, MEK, ERK, JNK and p38. The MAPKs are another important inflammatory mediators involved in the proliferation and migration of RA-FLS [50]. It has beens suggested that the enhanced anti-inflammatory effects of therapeutic inhibitors result from targeting the JNK and p38 MAPK pathways [50,51]. MAPKs signaling pathways regulate the production of chemokines, tissue destructive enzymes and many inflammatory cytokines in RA patients [50,51]. Therefore, the results of the current study indicate that hsa-miR-137 may regulate intra-articular inflammation by targeting the p38 MAPK signaling pathways.

We also found that PMs induce ROS production, which is associated with IL-6 and COX-II production. Serum level of ROS is effective as biomarkers for monitoring RA disease progression [52]. ROS can activate the signaling cascades of inflammatory cells to synthesize pro-inflammatory cytokines and we found that IL-6 and COX-II are the downstream mediators. Particulate matter has been found to induce ROS-associated signaling pathways and subsequent inflammation in various organ systems, including skin keratinocyte, brain nerve cells (SH-SY5Y) and cardiovascular system [53,54,55]. In this study, we demonstrated that PMs-induced ROS production in RA-FLS increases the expression of IL-6 and COX-II by downregulating hsa-miRNA-137.

## 5. Conclusions

We have shown that exposure to PMs results in hsa-miR-137 downregulation and an increase of IL-6 and COX-II production in RA-FLS, causing inflammation and RA exacerbation. The molecular mechanisms include ROS-mediated MAPK signaling pathways that cause increased IL-6 and COX-II through downregulation of hsa-miR-137 (Figure 7). A better understanding of the regulatory pathways helps to develop therapeutic strategies and potential targeting. In the future, further research is needed to investigate whether targeting hsa-miR-137 or inhibitors of ROS/MAPK signaling pathways can suppress the PMs-induced RA exacerbation.

## Figures and Tables

**Figure 1 cells-09-01378-f001:**
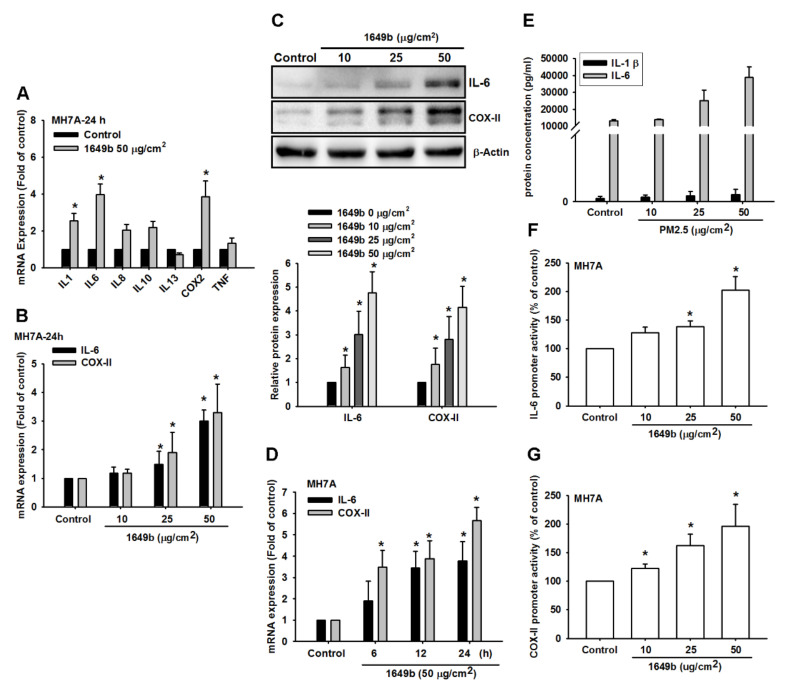
Concentration- and time-dependent increase in IL-6 production by particulate matter (PMs). (**A**) Human rheumatoid arthritis fibroblast-like synoviocytes (RA-FLS) were incubated with PMs (50 μg/cm^2^) for 24 h. The mRNA expression of inflammatory cytokines were examined using qPCR (*n* = 4); (**B**) RA-FLS were incubated with various concentrations of PMs for 24 h. The mRNA expression of IL-6 and COX-2 were examined using qPCR (*n* = 4); (**C**) RA-FLS were incubated with various concentrations of PMs for 24 h. The protein expression of IL-6 and COX-2 were examined using western blot (*n* = 4); (**D**) RA-FLS were incubated with PMs (50 μg/cm^2^) for 6, 12 or 24 h. The mRNA expression of IL-6 and COX-2 were examined using qPCR (*n* = 4); (**E**) RA-FLS were incubated with various concentrations of PMs for 24 h; supernatants and cell lysates were then collected. The IL-6 level in the culture media was measured using a Quantikine ELISA kit (*n* = 4); (**F**,**G**) RA-FLS were incubated with various concentrations of PMs for 24 h. IL-6 and COX-2 luciferase activity was measured and the results were normalized to the β-galactosidase activity. Results are expressed as mean ± SD. * *p* < 0.05 compared with the control; # *p* < 0.05 compared with the PM-treated group.

**Figure 2 cells-09-01378-f002:**
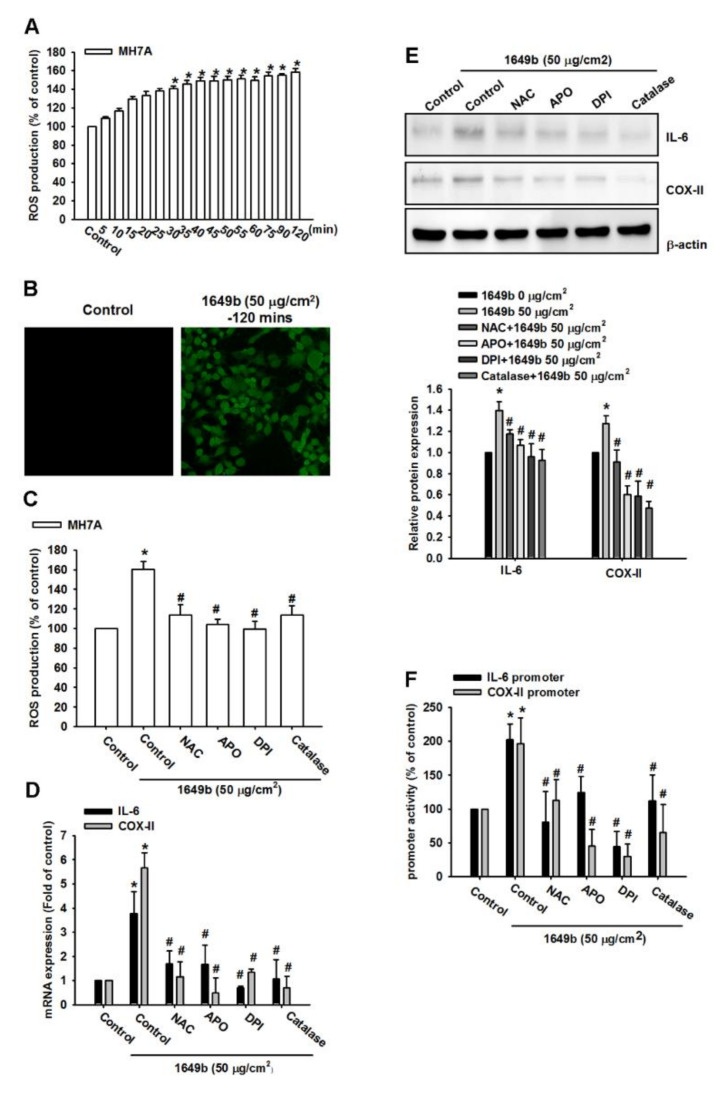
PMs induce the generation of reactive oxygen species (ROS) in human rheumatoid arthritis synovial fibroblasts (RASFs). (**A**,**B**) RA-FLS were treated with control solution or PMs (50 μg/cm^2^) for various times. The concentration of ROS production was examined by using flow cytometry and fluorescence image; (**C**) RA-FLS were pretreated for 30 min with N-acetylcysteine 1-mM (NAC, NADPH oxidase inhibitor), diphenyleneiodonium chloride 10 μM (DPI, nonspecific flavoprotein inhibitor), catalase 500 units/mL (ROS scavengers) and apocynin 100 μM (APO, NOX-like enzymes inhibitor) followed by treatment with PMs (50 μg/cm^2^) for 30 min. The concentration of ROS production was examined by using flow cytometry; (**D**–**F**) RA-FLS were treated with N-acetylcysteine 1-mM (NAC, NADPH oxidase inhibitor), diphenyleneiodonium chloride 10 μM (DPI, nonspecific flavoprotein inhibitor), catalase 500 units/mL (ROS scavengers) and apocynin 100 μM (APO, NOX-like enzymes inhibitor) followed by treatment with PMs (50 μg/cm^2^) for 24 h. The IL-6 and COX-2 expression were examined by using qPCR, western blotting and luciferase activity. The data are expressed as the mean ± SD. * *p* < 0.05 compared with controls. # *p* < 0.05 compared with the PMs treated groups.

**Figure 3 cells-09-01378-f003:**
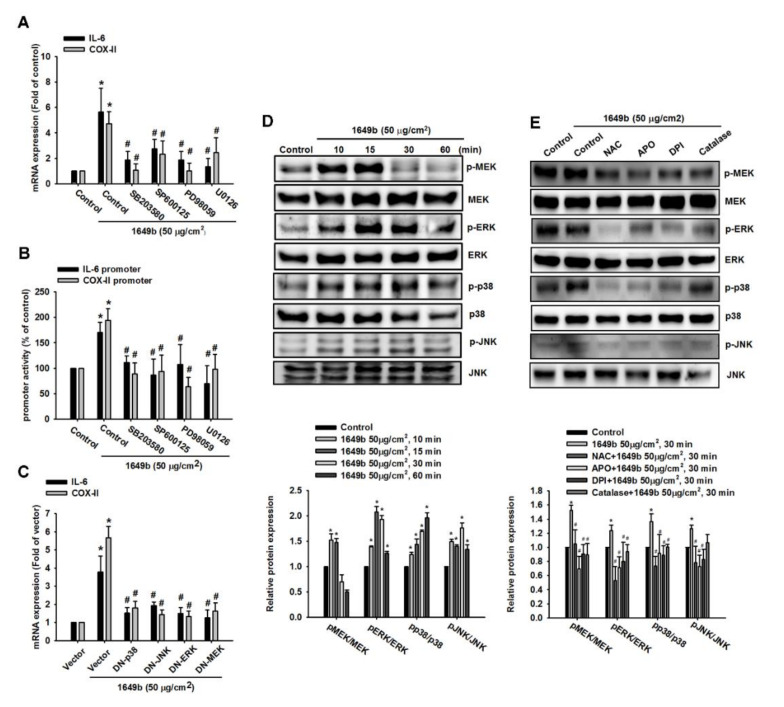
PMs-induced IL-6 and COX-2 expression via MAPK in human RA-FLS. (**A**,**B**) RA-FLS were first treated with U0126, PD98059, SB203580 and SP600125 and PMs stimulation, IL-6 and COX-II mRNA expression and luciferase activity were measured, respectively; (**C**) RA-FLS were transfected with MEK, ERK, p38 and JNK dominant negative (DN) mutants for 24 h and PMs stimulation, IL-6 and COX-2 mRNA were measured using qPCR; (**D**) At different PM stimulation durations (0, 10, 15, 30 and 60 min), MEK, ERK, p38 and JNK phosphorylated and total proteins were measured using Western immunoblotting; (**E**) RA-FLS were pretreated for 30 min with N-acetylcysteine (NAC, NADPH oxidase inhibitor), diphenyleneiodonium chloride (DPI, nonspecific flavoprotein inhibitor), catalase (ROS scavengers) and apocynin (APO, NOX-like enzymes inhibitor) followed by treatment with PMs (50 μg/cm^2^) for 15 min. MEK, ERK, p38 and JNK phosphorylated and total proteins were measured using Western immunoblotting. Results are expressed as mean ± SD, n = 4. * *p* < 0.05 compared with control; # *p* < 0.05 compared with the PM-treated group.

**Figure 4 cells-09-01378-f004:**
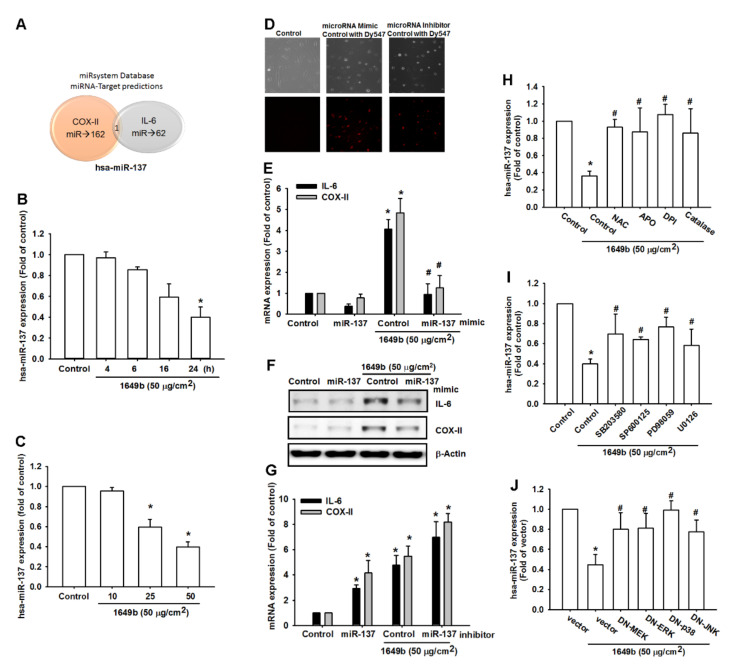
hsa-miR-137 inhibition is involved in PM-induced IL-6 and COX-2 expression. (**A**) Searches of the online computational algorithms (miRsystem) and miRNA-Target predictions for candidate miRNAs that target the IL-6 and COX-II-regions revealed the involvement of hsa-miR-137; (**B**,**C**) RA patient’s synovial fibroblast cells were incubated with various concentrations of PMs for 24 h or with PM2.5 (50 μg/cm^2^) for 4, 6, 16 or 24 h; hsa-miR-137 expression was assessed by qPCR; (**D**) Cells were transfected with fluorescent dye-labeled miRNA mimic control and miRNA inhibitor for 24 h, the fluorescence staining was examined by; (**E**–**G**) RASFs were transfected with hsa-miR-137 mimics or hsa-miR-137 inhibitors for 24 h, followed by stimulation with PMs for 24 h; IL-6 and COX-II-expression was examined by qPCR and western blot; (**H**–**J**) Cells were pretreated for 30 min with NAC, DPI, catalase and APO or MAPK inhibitor or MEK, ERK, JNK and p38 MAPK mutants stimulated with PMs for 24 h. hsa-miR-137 expression was assessed by qPCR. Results are expressed as the mean ± SD. * *p* < 0.05 as compared with baseline. # *p* < 0.05 as compared with the PMs-treated group.

**Figure 5 cells-09-01378-f005:**
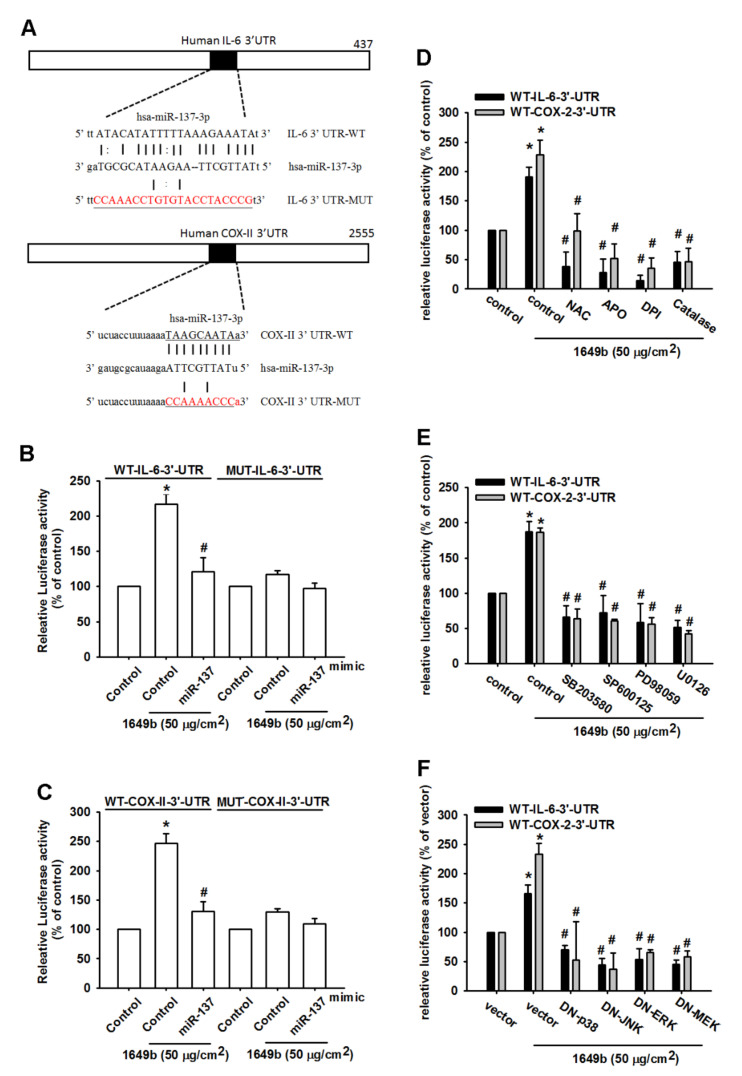
PMs increases IL-6 and COX-II-expression via inhibition of hsa-miR-137 through the MEK, ERK, p38 and JNK signaling pathways. (**A**) Schematic 3’-UTR representation of human IL-6 and COX-II containing the hsa-miR-137 binding site; (**B**,**C**) Cells were transfected with indicated luciferase plasmids before incubation with PM2.5 for 24 h; Luciferase activity was assessed; (**D**–**F**) Incubation with ROS, MEK, ERK, JNK and p38 inhibitors or MEK, ERK, JNK and p38 MAPK mutants reversed PM-mediated IL-6 and COX-II-3’-UTR luciferase activity. Results are expressed as the mean ± SD. * *p* < 0.05 as compared with baseline. # *p* < 0.05 as compared with the PM2.5-treated group.

**Figure 6 cells-09-01378-f006:**
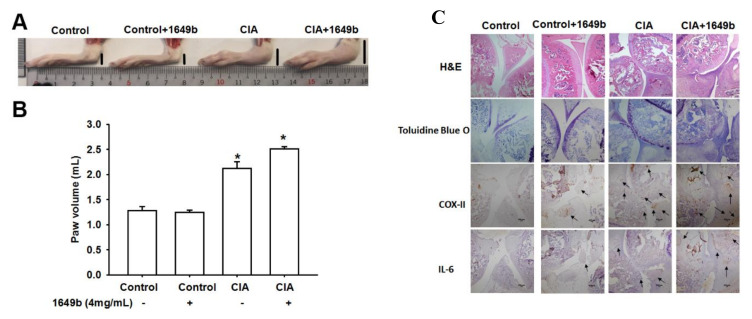
Administration of PMs augments bone erosion and IL-6 and COX-II expression in a collagen-induced arthritis (CIA) model. (**A**,**B**) Hind paw swelling was photographed and measured with a digital plethysmometer in healthy controls, untreated CIA mice and in CIA mice administered PMs 4 mg/kg (**C**) Histologic sections of ankle joints were stained with H&E and Toluidine blue O or immunostained with COX-II and IL-6 antibodies. Results are expressed as the mean ± SD. * *p* < 0.05 compared with controls; # *p* < 0.05 compared with the PMs-treated group.

**Figure 7 cells-09-01378-f007:**
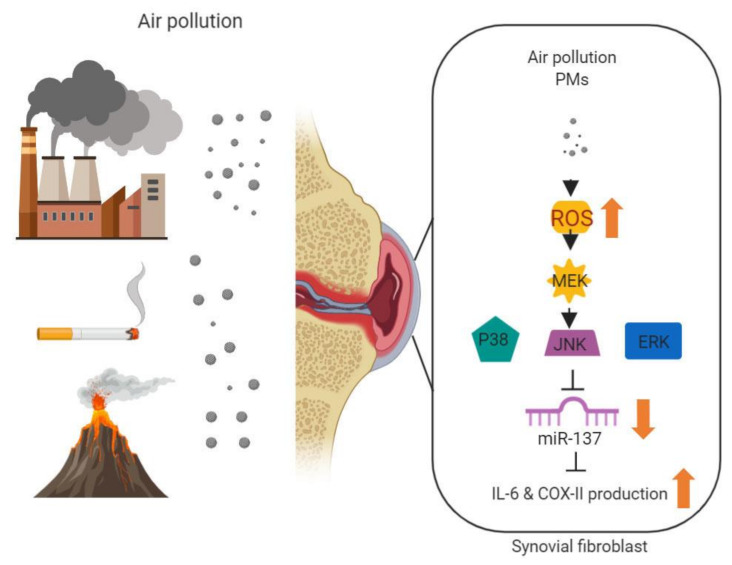
Schematic diagram illustrating the proposed signaling pathway involved in particulate matter (PM)-upregulation of IL-6 and COX-II expression via miR-137. PMs increase generation of reactive oxygen species (ROS) and in turn activate MAPK signaling pathways that cause increased IL-6 and COX-II through downregulation of miRNA-137.

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
