# Peer review of "Urban Particulate Matter Enhances ROS/IL-6/COX-II Production by Inhibiting MicroRNA-137 in Synovial Fibroblast of Rheumatoid Arthritis"

_cells, 2020, doi:10.3390/cells9061378_

Round 1

Reviewer 1 Report

  The authors aimed to clarify the role of PMs for exacerbation RA activity by molecular levels.  They demonstrated that PMs induce the IL-6 and COX-II production in RA-FLS and that ROS generation is necessary for IL-6 and COX-II expression.  Next, they also showed that MAPK involves the PMs-induced IL-6 and COX-II production.  Finally, they examined the effects of PMs using CIA model.  The author’s investigations are very interesting; however, I have some major and minor concerns. 

Major concerns:

1) The authors claimed that PMs enhance IL-6 and COX-II production in FLS with RA joints and summarized the mechanisms in Figure 7; however, PMs could affect alveolar epithelial cells in lungs with RA patients followed by  taking into the upper air tract.  It is needed to describe how PMs function in the joints through upper air tract in detail.  

2) In Figure 6, there is not Fig 6”C”.  In addition, they described that PMs augments disease activity in the CIA rat model.  However, there is not statistically significance in paw volume between CIA and CIA administrated PMs.  Also, in IHC staining, probably Figure 6C, we cannot find statistically differences between CIA and CIA administrated PMs.  These findings could not coincide with the fact that PMs enhanced arthritis.

Minor concerns:

1) In Material and Methods, they used human synovial tissues with RA patients.  It is needed to add their drugs including DMARDs or steroid.

2) Figure 1 legend is inserted into Results.

3) It is needed to check notation especially in the concentration of materials.

Author Response

RE: Cells-804637

Urban particulate matter enhances ROS/IL-6/COX-II production by inhibiting MicroRNA-137 in synovial fibroblast of rheumatoid arthritis

Dear Editor,

Thank you for your appreciated comments on our manuscript. We had the manuscript revised, all according to the reviewers’ and editor’s suggestions. We underline every change and highlight in red color on the revised manuscript. The replies for the reviewers’ criticisms are as followings. We hope this revised version can be acceptable.

Best regards,

Ming-Horng Tsai

Chief, Division of Neonatology and Pediatric Hematology/Oncology, Department of Pediatrics, Yunlin Chang Gung Memorial Hospital, Taiwan, R.O.C.

Comments from Reviewer No.1 :

Reviewer #1
The authors aimed to clarify the role of PMs for exacerbation RA activity by molecular levels. They demonstrated that PMs induce the IL-6 and COX-II production in RA-FLS and that ROS generation is necessary for IL-6 and COX-II expression. Next, they also showed that MAPK involves the PMs-induced IL-6 and COX-II production. Finally, they examined the effects of PMs using CIA model. The author’s investigations are very interesting; however, I have some major and minor concerns.

Major concerns:
1) The authors claimed that PMs enhance IL-6 and COX-II production in FLS with RA joints and summarized the mechanisms in Figure 7; however, PMs could affect alveolar epithelial cells in lungs with RA patients followed by taking into the upper air tract. It is needed to describe how PMs function in the joints through upper air tract in detail.
Ans: We agree reviewer's valid points and thank reviewer's suggestion.

   Air particulate matter according to size, grouping them as coarse, fine and ultrafine. Coarse particles fall between 2.5 microns and 10 microns in diameter and are called PM 10-2.5. Fine particles are 2.5 microns in diameter or smaller and are called PM2.5. Ultrafine particles are smaller than 0.1 micron in diameter1 and are small enough to pass through the lung tissue into the blood stream, circulating like the oxygen molecules themselves. In addition, accumulating evidence suggests that the associations between the incidence of RA and air pollution. However, the detail mechanism of PM2.5 in RA are still uncertain. Here, we found that PM2.5 promotes inflammatory cytokine IL-6 and COX-II production. PMs increase generation of reactive oxygen species (ROS), and in turn activate MAPK signaling pathways that cause increased IL-6 and COX-II through downregulation of miRNA-137. Our finding provides new insight for understanding the PM2.5 in RA.

2) In Figure 6, there is not Fig 6”C”. In addition, they described that PMs augments disease activity in the CIA rat model. However, there is not statistically significance in paw volume between CIA and CIA administrated PMs. Also, in IHC staining, probably Figure 6C, we cannot find statistically differences between CIA and CIA administrated PMs. These findings could not coincide with the fact that PMs enhanced arthritis.
Ans: We agree reviewer's valid points and thank reviewer's suggestion. The data have been checked and updated in Fig 6C. IHC staining revealed substantially higher IL-6 and COX-II expression in CIA rats compared with controls. Additionally, treatment with PMs also enhanced IL-6 and COX-II expression over synovium specimens (Fig. 6C). These results indicated that PMs augments disease activity in the CIA rat model.

Minor concerns:
1) In Material and Methods, they used human synovial tissues with RA patients. It is needed to add their drugs including DMARDs or steroid.

Ans: Thank you for your instructive advice. Among the 12 RA patients, 7 (58.3%) were under steroid treatment, and all patients had DMARDs at the time of enrollment. The DMARDs included methotrexate (7 patients, 58.3%), hydroxychloroquine (3 patients, 25%), and sulfasalazine (2 patients, 16.7%) (Line 127)

2) Figure 1 legend is inserted into Results.

Ans: We agree reviewer's valid points and thank reviewer's suggestion. The Figure 1 legend is inserted into Results at line 237-256.

3) It is needed to check notation especially in the concentration of materials.
Ans: We agree reviewer's valid points and thank reviewer's suggestion. It has been corrected in the revised manuscript.

Reviewer 2 Report

The authors demostrated that PM increased the level of IL-6 and COX2 via the downregulation of microRNA-137. The manuscript is easy to read and well written. However, some points should be clarified.

  1. Both MH7A and normal primary RA-FLS (H-FLS) were used for experiments. H-FLS is very limited to grow. I understand it. In Figure legend, which cells MH7A or H-FLS were used should be indicated clearly. Some figures are confused to know which cell were used.
  2. PM treatments concentration was 10-50 ug/cm2. As you know, air pollution uses PM ug/m3 unit. Even though it is in vitro experiments, the concentration is too high to demonstrate the real effect of PM on inductin of RA.
  3. In rat experiments, the authors may get more information but shows a little results. For example, serum cytokine (IL6 or inflammatory cytokines) level or serum miRNA-137 level should be checked. However, the authors showed only paw volume oand histology. This part should more detailed described. It is important part to prove the effect of PMs.
  4. For in vivo experimens, PMs 4 mg/kg was treated. Please show are the reference or preliminary results about why the conetration were treated.
  5. CIA was induced. The effect of CIA-induced inflammation induction is so strong that PM-iflammation effects would be overwhelmed. I think that it is amzing that PM treatment induced more paw volume. What do you think of it?

Author Response

RE: Cells-804637

Urban particulate matter enhances ROS/IL-6/COX-II production by inhibiting MicroRNA-137 in synovial fibroblast of rheumatoid arthritis

Dear Editor,

Thank you for your appreciated comments on our manuscript. We had the manuscript revised, all according to the reviewers’ and editor’s suggestions. We underline every change and highlight in red color on the revised manuscript. The replies for the reviewers’ criticisms are as followings. We hope this revised version can be acceptable.

Best regards,

Ming-Horng Tsai

Chief, Division of Neonatology and Pediatric Hematology/Oncology, Department of Pediatrics, Yunlin Chang Gung Memorial Hospital, Taiwan, R.O.C.

Comments from Reviewer No.2 :

Reviewer #2
The authors demonstrated that PM increased the level of IL-6 and COX2 via the downregulation of microRNA-137. The manuscript is easy to read and well written. However, some points should be clarified.

1. Both MH7A and normal primary RA-FLS (H-FLS) were used for experiments. H-FLS is very limited to grow. I understand it. In Figure legend, which cells MH7A or H-FLS were used should be indicated clearly. Some figures are confused to know which cell were used.

Ans: We thank the editor for your comments. It has been corrected in the revised manuscript. We use RA patient’s synovial fibroblast cell for Fig 4B-C. It has been corrected in the revised manuscript.

  1. PM treatments concentration was 10-50 ug/cm2. As you know, air pollution uses PM ug/m3 unit. Even though it is in vitro experiments, the concentration is too high to demonstrate the real effect of PM on inductin of RA.

Ans: We agree reviewer's valid points and thank reviewer's suggestion. In the experimental design of the current study may not fully reflect the real condition of PM on the RA induction, including the exposure period and doses. This is a experimental cell model to response the possibility of PM exposure on RA. The range of PM doses were used in several studies [1]. And according our previous experiences for PM on the in vitro studies [2], the PM doses on the cell, 25 μg/cm2 is lower, and 75μg/cm2 is higher but not caused larger cells apoptosis, and the time 24 hour could be considered as an enough time for cell to be induced inflammatory signals.

1.Cho CC, Hsieh WY, Tsai CH, Chen CY, Chang HF, Lin CS. (2018) In Vitro and In Vivo Experimental Studies of PM2.5 on Disease Progression. Int J Environ Res Public Health. 15(7). pii: E1380. doi: 10.3390/ijerph15071380.

2.Lee, C.W., Lin, Z.C., Hu, S.C.S., Chiang Y.C., Hsu, L.F., Lin, Y.C., Lee, I.T., Tsai, M.H., Fang, J.Y. (2016) Urban particulate matter down-regulates filaggrin via COX2 expression /PGE2 production leading to skin barrier dysfunction. Sci. Rep. 6:27995.

  1. In rat experiments, the authors may get more information but shows a little results. For example, serum cytokine (IL6 or inflammatory cytokines) level or serum miRNA-137 level should be checked. However, the authors showed only paw volume and histology. This part should more detailed described. It is important part to prove the effect of PMs.

Ans: We agree reviewer's valid points and thank reviewer's suggestion. We are so sorry about that we can’t provid the serum cytokine (IL6 or inflammatory cytokines) level or serum miRNA-137 level in rat experiments, because we didn’t keep the serum from the experimental rats.

   A series of studies suggested that miR-137 played important functional roles in the development of rheumatoid arthritis. The study by Zhang et al. found that IL-1β decreased the expression of miR-137 in the chondrocytes and the miR-137 expression level was lower in the OA cases than in the control patients. In addition, it was been reported from 2018 by Du et al. showed that miR‑137 may serve an inhibitory role in RA by targeting CXCL12 expression, and miR‑137 may be a potential target for the treatment of RA. Therefore, this study aimed to determine whether miR‑137 effected inflammation in RA‑FLS, and the results revealed that the overexpression of miR‑137 substantially decreased the expression of IL‑6 and COX-II in RA‑FLS, suggesting that miR‑137 may have potential as an anticytokine therapy for RA.

  1. For in vivo experiments, PMs 4 mg/kg was treated. Please show which are the reference or preliminary results about why the concentration were treated.

Ans: We've added the references of the PM concentration we selected (as showed below), and provide more information of PM (SRM-1649b) in the revision of manuscript. Moreover, in our previous study also showed the severity of osteoarthritis could be promoted by PM exposure with a PM concentration effect (5 mg/kg has significant effects than 2 and 1 mg/kg) [6].

1.Pei, Y., Jiang, R., Zou, Y., Wang, Y., Zhang, S., Wang, G., et al. (2016) Effects of Fine Particulate Matter (PM2.5) on Systemic Oxidative Stress and Cardiac Function in ApoE(-/-) Mice. Int J Environ Res Public Health. 13 5, https://doi.org/10.3390/ijerph13050484

2.Liu, Y., Wang, L., Wang, F., Li, C. (2016) Effect of Fine Particulate Matter (PM2.5) on Rat Placenta Pathology and Perinatal Outcomes. Med Sci Monit. 22:3274-3280

3.Luo, B., Shi, H., Wang, L., Shi, Y., Wang, C., Yang, J., et al. (2014) Rat lung response to PM2.5 exposure under different cold stresses. Int J Environ Res Public Health. 11 12:12915-12926, https://doi.org/10.3390/ijerph111212915

4.Gerlofs-Nijland, M.E., Boere, A.J., Leseman, D.L., Dormans, J.A., Sandstrom, T., Salonen, R.O., et al. (2005) Effects of particulate matter on the pulmonary and vascular system: time course in spontaneously hypertensive rats. Particle and fibre toxicology. 2 1:2, https://doi.org/10.1186/1743-8977-2-2

5.Wang, H., Song, L., Ju, W., Wang, X., Dong, L., Zhang, Y., et al. (2017) The acute airway inflammation induced by PM2.5 exposure and the treatment of essential oils in Balb/c mice. Scientific reports. 7:44256, https://doi.org/10.1038/srep44256

6.Peng KT, Liu JF, Chiang YC, Chen PC, Chiang MH, Shih HN, Chang PJ, Lee CW (2019) Particulate matter exposure aggravates osteoarthritis severity. Clin Sci (Lond). 133(21):2171-2187. doi: 10.1042/CS20190458.

  1. CIA was induced. The effect of CIA-induced inflammation induction is so strong that PM-inflammation effects would be overwhelmed. I think that it is amazing that PM treatment induced more paw volume. What do you think of it?

Ans: We agree reviewer's valid points and thank reviewer's suggestion.

Rheumatoid arthritis (RA) is an autoimmune disease. Although the joints are the major targets of RA, lung also can be involved. Lung disease is common in RA, Vital capacity diminishes, bronchiectasis develops, and an infiltrate of lymphocytes and plasma cells indicates chronic inflammation [1]. Diffuse idiopathic interstitial lung disease is found in nearly 60% of patients with established RA. The posited involvement of the lung in the pathophysiology of RA has received support over the last decade from studies into the effects on the disease of several inhaled environmental factors. Smoking is strongly associated with the development, severity, and treatment response of RA [2]. In addition, Silica, which is also found in ambient air, has been incriminated in the pathophysiology of RA [3, 4]. Therefore, air pollution is a focus of research in the field of autoimmunity.

   In this paper, we have shown that exposure to PMs result in hsa-miR-137 downregulation and an increase of IL-6 and COX-II production in RA-FLS, causing inflammation and RA exacerbation. The molecular mechanisms include ROS-mediated MAPK signaling pathways that cause increased IL-6 and COX-II through downregulation of hsa-miR-137 (Figure 7). A better understanding of the regulatory pathways helps to develop therapeutic strategies and potential targeting. In the future, further research is needed to investigate whether targeting hsa-miR-137 or inhibitors of ROS/MAPK signaling pathways can suppress the PMs-induced RA exacerbation.

  1. Fischer, A., et al., Lung disease with anti-CCP antibodies but not rheumatoid arthritis or connective tissue disease. Respir Med, 2012. 106(7): p. 1040-7.
  2. Klareskog, L., et al., A new model for an etiology of rheumatoid arthritis: smoking may trigger HLA-DR (shared epitope)-restricted immune reactions to autoantigens modified by citrullination. Arthritis Rheum, 2006. 54(1): p. 38-46.
  3. Speck-Hernandez, C.A. and G. Montoya-Ortiz, Silicon, a Possible Link between Environmental Exposure and Autoimmune Diseases: The Case of Rheumatoid Arthritis. Arthritis, 2012. 2012: p. 604187.
  4. Caplan, A., Certain unusual radiological appearances in the chest of coal-miners suffering from rheumatoid arthritis. Thorax, 1953. 8(1): p. 29-37.

Round 2

Reviewer 2 Report

The manuscript was edited properly according to the comments.